# Structural Variants and Implicated Processes Associated with Familial Tourette Syndrome

**DOI:** 10.3390/ijms25115758

**Published:** 2024-05-25

**Authors:** Jakub P. Fichna, Mateusz Chiliński, Anup Kumar Halder, Paweł Cięszczyk, Dariusz Plewczynski, Cezary Żekanowski, Piotr Janik

**Affiliations:** 1Department of Neurogenetics and Functional Genomics, Mossakowski Medical Research Institute, Polish Academy of Sciences, 02-106 Warsaw, Poland; c.zekanowski@imdik.pan.pl; 2Laboratory of Bioinformatics and Computational Genomics, Faculty of Mathematics and Information Science, Warsaw University of Technology, 00-662 Warsaw, Poland or m.chilinski@cent.uw.edu.pl (M.C.); or a.halder@cent.uw.edu.pl (A.K.H.); or d.plewczynski@cent.uw.edu.pl (D.P.); 3Laboratory of Functional and Structural Genomics, Centre of New Technologies, University of Warsaw, 02-097 Warsaw, Poland; 4Section for Computational and RNA Biology, Department of Biology, University of Copenhagen, 2200 Copenhagen, Denmark; 5Faculty of Physical Education, Gdansk University of Physical Education and Sport, Górskiego 1 Street, 80-336 Gdansk, Poland; pawel.cieszczyk@awf.gda.pl; 6Department of Neurology, Medical University of Warsaw, 02-091 Warsaw, Poland; piotr.janik@wum.edu.pl

**Keywords:** tic disorders, copy number, CNV, duplication, inversion, polygenic, neurexin, variant burden, schizophrenia, misophonia

## Abstract

Gilles de la Tourette syndrome (GTS) is a neurodevelopmental psychiatric disorder with complex and elusive etiology with a significant role of genetic factors. The aim of this study was to identify structural variants that could be associated with familial GTS. The study group comprised 17 multiplex families with 80 patients. Structural variants were identified from whole-genome sequencing data and followed by co-segregation and bioinformatic analyses. The localization of these variants was used to select candidate genes and create gene sets, which were subsequently processed in gene ontology and pathway enrichment analysis. Seventy putative pathogenic variants shared among affected individuals within one family but not present in the control group were identified. Only four private or rare deletions were exonic in *LDLRAD4*, *B2M*, *USH2A*, and *ZNF765* genes. Notably, the *USH2A* gene is involved in cochlear development and sensory perception of sound, a process that was associated previously with familial GTS. In addition, two rare variants and three not present in the control group were co-segregating with the disease in two families, and uncommon insertions in *GOLM1* and *DISC1* were co-segregating in three families each. Enrichment analysis showed that identified structural variants affected synaptic vesicle endocytosis, cell leading-edge organization, and signaling for neurite outgrowth. The results further support the involvement of the regulation of neurotransmission, neuronal migration, and sound-sensing in GTS.

## 1. Introduction

Gilles de la Tourette syndrome (GTS) is a developmental neuropsychiatric disorder characterized by a combination of motor and vocal/phonic tics. The tics have pre-pubertal onset with common significant improvement during adolescence. Many GTS patients present a plethora of additional psychiatric comorbidities, further suggesting a complex etiology [1,2]. The phenotype may be influenced by numerous factors, including environmental, prenatal and perinatal ones, hormonal disturbances, as well as psychosocial stressors [3,4,5,6]. The clinical phenotype of GTS belongs to a broader entity of tic disorders (TDs) [7]. The prevalence of GTS in the general pediatric population ranges from 0.3% to 0.77% and is 10–100 fold lower in adulthood, but TDs are more common affecting up to 5% of the general population [2,8].

The heritability of GTS and TDs is estimated to be 60–80% [1,9]. Candidate gene approach and linkage studies indicated multiple genes as possibly important in GTS etiology [10,11]. The protein products of these genes are usually involved in neurotransmitter signaling; synapse development, organization, and function; axon differentiation; cell adhesion; and mitochondrial activity [11,12]. However, only one variant in *HDC* and one in *SLITRK1* are recognized by Human Phenotype Ontology as pathogenic in GTS based mainly on co-segregation studies in one family each [13].

Recent next-generation sequencing studies revealed a complex genetic background of GTS involving multiple interacting genes [14] and a role of the rare variant burden in TDs [15]. The available data suggest that de novo variants in approximately 400 genes contribute to GTS risk in 12% of clinical cases [16]. It is increasingly evident that rare pathogenic single-nucleotide variants in a single gene cannot be responsible for a substantial fraction of GTS cases.

About 21% of the heritability of GTS can be explained by polymorphisms with an MAF (minor allele frequency) between 0.1 and 5% [5]. Other GWAS studies have shown that GTS has an overlapping and polygenic background with autism spectrum disorder (ASD), obsessive compulsive disorder (OCD), attention deficit hyperactivity disorder (ADHD), and major depressive disorder (MDD) [17]. Also, low-impact common variants have been proposed to contribute to the GTS phenotype in a polygenic manner [18]. Epigenetic mechanisms, including functional networks of non-coding RNAs (ncRNAs), have also been proposed to mediate the influence of environmental factors on the genetic background of GTS. Also, the role of rare non-coding variants remains largely unexplored in TDs. All of this makes the identification of GTS susceptibility genes a complex and challenging task.

Additional components that constitute the genomic background of the GTS are structural variants (SV), including short insertions, and deletions (indels). The genomic difference between individuals associated with SVs is 3–10 times higher than that caused by single-nucleotide variants (SNVs) [19]. As a result, SVs may have greater impacts on gene functions than SNVs. Until now, SVs have been linked to a number of human diseases, including neurodevelopmental disorders [20]. SVs can be detected using either array-based detection (e.g., array CGH) or sequencing-based computational methods, which have higher sensitivity for small SVs. The problem with identifying SVs is the inability of a single computational algorithm to accurately and sensitively detect all types and sizes of SVs. The solution is to use a pipeline that combines several algorithms to call SVs and then combine the results.

Here, we hypothesize that familial GTS cases in the Polish population are best explained by an oligogenic model of inheritance, with multiple variants, including structural variants.

To verify this hypothesis, we analyzed a group of 17 families with 80 patients and 44 healthy family members using whole-genome sequencing (WGS) to identify structural variants associated with GTS. Overall, the results confirm a substantial contribution of SVs in numerous genes to the genomic background of GTS in Polish families. Enrichment analysis of the variants revealed their association with known processes involved in the etiology of GTS, as well as with novel processes not previously implicated in the disease.

## 2. Results

The analysis revealed 795844 structural variants, an average of 2726 per genome, including 5385 unique variants present in all the patients from any one family (Appendix A). Most of these variants were found in non-coding regions.

### 2.1. Rare Variant Analysis

Analysis of variants segregating according to the applied schemes in at least one family revealed 211 that were uncommon (<5%), including 97 rare (<1%), and 70 that were absent (0%) in the control group. Most of them were deletions (154 uncommon, 70 rare, and 53 private), followed by duplications (34, 14, and 9, respectively), inversions (11, 4, and 2, respectively), and insertions (7, 3, and 2, respectively). None of the variants not present in the control group were found in the GnomAD SVs v4 database.

Only four of the variants that were either rare or private, i.e., not present in the control group, were at least partly exonic, with two affecting coding sequences (in LDLRAD4 and B2M) and two in 3′UTRs (in USH2A and ZNF765) (Table 1).

Thirteen structural variants that were uncommon in the control group were co-segregating with a disease in two families, with three variants co-segregating in three families (Table 2). Eight of these sixteen variants were located in a gene (*SERF2*, *ARHGEF4*, *GOLM1*, *MSI1*, *ZNF469*, *DISC1*, *CCPG1*, *ENSG00000258464*), although all of them were intronic. *ZNF469* was the only gene with two different variants co-segregating with a disease, with one duplication present in two families and one insertion present in a third family.

### 2.2. Overlap with a Single-Nucleotide Variant Analysis

Comparison with our previous analyses [21] on single-nucleotide variants revealed some overlap.

Three genes (*NRXN3*, *PSD3*, and *ZNF407*) had private structural variants co-segregating with a disease in one family and very rare (<0.1%) single-nucleotide variants co-segregating with a disease in at least two families.

Three genes (*EYS*, *NRXN3*, and *ZNF407*) had uncommon structural variants co-segregating with a disease in one family and rare (<1%) single-nucleotide variants co-segregating with a disease in at least three families.

Four genes (*EYS*, *NRXN3*, *CACNA2D3*, and *USH2A*) had uncommon structural variants co-segregating with a disease in one family and uncommon (<5%) single-nucleotide variants co-segregating with a disease in at least four families.

### 2.3. Enrichment Analysis

All genes identified in rare structural variant co-segregation studies, and separately, the above-mentioned 15 genes, including the overlap of structural variant and single-nucleotide variant co-segregation, were analyzed using Metascape software, which showed a significant enrichment of variants in genes involved in biological processes (Table 3 and Table 4).

## 3. Discussion

To gain an insight into the contribution of structural variants to the genomic basis of Gilles de la Tourette syndrome, we examined seventeen large GTS-risk families from the Polish population, using whole-genome sequencing, structural variant detection, and subsequent bioinformatic analyses. We have also overlaid the results with our previous studies of single-nucleotide variants. We posit that genes with rare variants co-segregating with the disease are likely to play a role in the disease etiology. The majority of the structural variants found are intronic or intergenic. These variants may still affect the gene in which they reside altering splicing, or through exonisation [22,23], and may also modify regulatory processes, for example, by modulating non-coding RNAs or their targets [24].

### 3.1. Rare Variant Analysis

Four rare (<1% in the control group) exonic SV, in *LDLRAD4*, *B2M* (each encompassing coding exon), *ZNF765*, and *USH2A* (each in the 3UTR), were co-segregating with the disease in one family each. Although variation in these genes has not been linked to GTS to date, some variants, including structural variants, have been reported in other neuropsychiatric developmental disorders. In addition, their involvement in neurological processes may be relevant to the pathology of the disease.

A deletion of 491 bp in exon 7 of the *LDLRAD4* gene was found in all three patients of family J, one of the smallest families in the study group (Table 5). The variant was also found in one of the 102 control subjects. LDLRAD4 is a low-density lipoprotein receptor class A domain-containing 4 protein highly expressed in the brain. It is a negative regulator of the TGF-beta signaling pathway, which regulates the growth, proliferation, differentiation, apoptosis, cell migration, and matrix protein production [25]. Deletions of 18p11.21 encompassing the *LDLRAD4* gene have previously been found in ASD and ADHD cases [26], which are GTS comorbid disorders.

A deletion covering part of exon 2 and the whole exon 3 of the *B2M* gene was found in all six affected members of family Y but was not present in any of the control subjects nor members of the other families. Interestingly, an apparently linked intronic deletion located 4bp upstream was found in the same genomes. *B2M*, sometimes used as a reference gene in expression studies, encodes beta-2-microglobulin, a serum protein component of the major histocompatibility complex (MHC) class I heavy chain on all nucleated cells. The main physiological function of MHC is to present the foreign antigens to autologic Lymphocyte T, which is important in initiating the immunological response. Pathogenic variants of the *B2M* gene may result in an abnormal immunological response, which is in line with the hypothesis that immunological abnormalities could play a role in GTS etiology [27,28]. In addition, MHC-I is necessary for neurite outgrowth and elongation, and neuronal polarization [29]. Beta2-microglobulin is involved in the negative regulation of forebrain neuron differentiation, the process that stops, prevents, or reduces the frequency, rate, or extent of forebrain neuron differentiation. Morphology and function of the prefrontal and frontal cortex are altered in GTS [30]. Silencing *B2M* results in astrocytic hypertrophy and fewer complex cell projections, suggesting the importance of MHC-I for synaptic stability in the brain [31]. Pathogenic single-nucleotide variants in *B2M* have been found in immunodeficiency [32] and in visceral amyloidosis with polyneuropathy [33]. MHC- I and -II alleles were found to influence the risk of childhood-onset OCD [34], and *B2M* was recently found to be significantly upregulated in chronic OCD cases [35].

A deletion of more than 1 kb in the non-coding exon 4 located in the 3′UTR of the *ZNF765* gene was found only in members of family I (four patients and one asymptomatic carrier) but was not present in any other familial samples nor healthy controls. The main function of zinc finger proteins (ZNFs) is regulating gene transcription through its nucleic acid-binding capability. Zinc finger protein 765 is a zinc finger protein related to the permeability of the blood–tumor barrier. A deletion encompassing *ZNF765* has already been described in a family with OCD, in all affected individuals with the disease, but also in a healthy carrier [36] (https://tspace.library.utoronto.ca/bitstream/1807/68833/1/Rajendram_Rageen_201411_MSc_thesis.pdf accessed on 20 February 2024).

A deletion of 77 bp in the non-coding exon 72 in the 3′UTR of the *USH2A* gene was found in four patients from family T with paternal inheritance. The variant was also found in one of 102 control subjects. The *USH2A* gene encodes a transmembrane usherin expressed in various tissues, including cochlear hair cells. It plays a crucial role in cochlear development and consequently, the perceptiod5z4 bgbvgcn of sound, and therefore *USH2A* mutations can result in the hearing impairment phenotype of Usher syndrome. Moreover, a deletion encompassing the *USH2A* gene was found in a patient with autism and speech delay [37]. Patients with neurodevelopmental disorders, including GTS and misophonia or auditory hypersensitivity, have already been reported, and enrichment of single-nucleotide variants in genes associated with sound processing by cells of the cochlea has already been found in GTS [21]. In rat model of GTS, a drastic decrease in *USH2A* methylation level was found [38].

Three structural variants were found co-segregating with the disease in three families: one intergenic insertion, one insertion in intron 3 *GOLM1* and one insertion in intron 3 of *DISC1* gene. *DISC1* variants were found in other psychiatric disorders, which was emphasized by its name derived from “disrupted in schizophrenia” [39]. Its protein product is involved in pyramidal neuron migration to cerebral cortex, and also plays a role in forebrain neuron differentiation and neuron development [40]. These processes have already been suggested as important in the etiology of GTS.

### 3.2. Overlap with a Single-Nucleotide Variant Analysis Results

In six genes (*NRXN3*, *PSD3*, *ZNF407*, *EYS*, *CACNA2D3* and *USH2A*), both single-nucleotide and structural variants were co-segregated with the disease.

Neurexins are cell-surface receptors for neuroligins to form a transsynaptic complex involved in synaptic junction and neurotransmission that have often been reported as implicated in neuropsychiatric disorders, especially ASD [41]. The phenotype of patients with neurexin-III deficiency may include developmental delay or learning impairment, movement disorder, and behavioral problems [42]. *NRXN3* has already been proposed as a candidate gene in GTS linkage studies [43], and its deletions have been found in ASD [44]. Deletions in another neuroligin gene, *NRXN1*, have been reported in GTS [45]. Furthermore, a familial linkage study pinpointed a locus that includes *NLGN1* [46].

CACNA2D3 protein plays an essential role in sensory filtering and habituation, a process of ignoring unimportant stimuli, impaired in many neuropsychiatric disorders (ADHD, schizophrenia, and autism) [47]. A leucine-rich transmembrane domain embedded in *CACNA2D3* together with neurexins and neuroligins has already been suggested in a pathogenetic model of GTS [48]. Disrupting splice site mutation in *CACNA2D3* has been found in autism spectrum disorder [49]. Mice with *CACNA2D* deletions showed ASD-like phenotype: anxiety, impaired sociability, increased repetitive behavior [50], and impaired sensory processing through neuronal deficits in the auditory and the acoustic startle pathway [51].

Interestingly, no other rare structural variant shared by all patients within any family in our study was located in genes whose protein products participate in neurotransmitter pathways of the cortex–striatum–thalamus–cortex (CBGTC) loops, which have often been the primary source of candidate genes in GTS studies.

Variants in *PSD3* have been found, among others, in variant burden polygenic studies of both GTS and ADHD [15,52]. Structural and missense variants in *ZNF407* have been found in autism [53], OCD, and schizophrenia [54]. Deletion of EYS has been found in dyslexia, but with incomplete penetrance [55].

### 3.3. Enrichment Analysis

Analysis of the genes with detected SVs revealed their association with various processes, pathways, and structures, including leading edge membrane and regulation of binding. These may be related to the etiology of GTS.

Many of the genes with structural variants co-segregating with familial GTS are involved in neurological processes, as indicated by the enrichment analysis, which showed that four out of the nine most significantly enriched processes (Table 2) are specifically related to the nervous system.

Synaptic vesicle endocytosis, a process enriched with structural variants, is crucial for synaptic transmission and, therefore, for the proper functioning of the nervous system. Neurons can maintain a high rate of neurotransmitter release only by recycling synaptic vesicle membranes through endocytosis. The process must be highly efficient to allow thousands of exo–endocytic cycles, and any impairment of this process could result in neurological disorders. Deletion covering *STON2*, which encodes the endocytic adaptor stonin 2, has been already found in familial GTS [43], and polymorphisms have been associated with schizophrenia [56]. Loss of this protein in mice causes increased locomotion and exploration-related behavior resembling the arousal-seeking and impulsivity observed in Tourette syndrome or schizophrenia [57].

Neurite outgrowth, a process where neurons create new projections, is fundamental in the differentiation of neurons. The migratory capacity of neuroblasts and axons is crucial in neurodevelopment and neuroregeneration [58]. Migration of any motile cells is possible due to cytoskeleton remodeling and adhesion dynamics. The plasma membrane surrounding the leading edge, with its ability to convert its tension into a mechanical signal affecting the lamellipodium, plays a vital role in cell migration. Abnormal neuronal migration has already been suggested in GTS with the observation of altered neuron numbers and distribution in the basal ganglia and striatum [59,60,61]. Impaired neuronal migration during brain development has also been proposed as a possible pathophysiology of other neurodevelopmental disorders, such as autism [62]. Synaptic migration and reorganization is also important in auditory perception as it plays a role in cochlear development and recovery after noise exposure [63]. Misophonia and sound hypersensitivity have already been reported in patients with neuropsychiatric disorders, including GTS [64,65,66,67].

### 3.4. Limitations

Most of the structural variants identified are located in regions that do not code protein products, either intronic or even intergenic; therefore, it is difficult to assess their impact on gene expression. Additional functional studies will be needed to determine the effects of individual variants. The putative causal association with the disease was based mainly on the co-segregation with the clinical phenotype, and additionally on the rarity of the variants. The frequency of SVs is much harder to asses than SNVs, as the gnomad SV database was created on smaller numbers of genomes than the gnomad SNV database. In addition, the method of SV detection used in gnomAD was different, although rather more sensitive and less specific.

We identified enriched processes and functions using databases, which assign genes to particular terms based on current knowledge and computational predictions.

The aim of our research is neither functional nor translational, and further work is needed to understand the genetic contribution to the clinical phenotype of GTS and the underlying dysfunctions at the molecular level.

## 4. Materials and Methods

### 4.1. Patients

The patient group was recruited as described earlier [21]. In brief, all the patients were diagnosed according to the Diagnostic and Statistical Manual of Mental Disorders criteria (DSM-IV-TR, DSM-5), using a semi-structured interview based on the TIC (Tourette syndrome International database Consortium) Data Entry Form [68]. The prevalence of the most common comorbid disorders was evaluated in psychiatric clinics.

All the patients were Caucasian from the ethnically homogeneous Polish population [69].

In case of a positive family history, DNA was collected from all available affected relatives and healthy members of the proband’s family.

Each patient was assigned to one of the following groups: GTS, non-GTS TDs, or healthy controls. The study group comprised 17 multiplex families with 80 patients (40 with GTS and 40 with non-GTS TDs) and 44 healthy family members (Table 5 and Appendix A). The control group consisted of 102 Polish athletes [70].

### 4.2. Whole-Genome Sequencing

Genomic DNA was extracted from peripheral blood leukocytes [71], and whole-genome sequencing (WGS) was performed by Novogene (Beijing, China) according to the previously used protocol [21]. 

### 4.3. Structural Variant Analysis

The structural variants were identified using ConsensusSV pipeline (https://github.com/SFGLab/ConsensuSV-pipeline accessed on 10 March 2024). The samples were first controlled using FastQC (http://www.bioinformatics.babraham.ac.uk/projects/fastqc accessed on 17 July 2023) and then aligned to the reference genome using bwa-mem algorithm. The output was converted to BAM format, sorted and indexed using samtools. Duplicates were marked using bammark duplicates from Picard (Picard Tools version 3.1.1) and then GATK (Van der Auwera and O’Connor, 2020) [72], with best practices of base recalibration and BQSR application performed. Single-nucleotide variants (SNPs) and indels were called with the bcftools [73]. For structural variants, ConsensusSV pipeline used merged callings from BreakDancer, BreakSeq, CNVnator, Delly, Lumpy, Manta, Tardis, and Whamg tools [74]. The final output was in the form of three VCF files for each sample—one with SNPs, one with indels, and one with structural variants. The confirmation of the identified variants was performed by visual examination of aligned sequences (bam files) in the Integrative Genomics Viewer software version 2.17.0.

### 4.4. Co-Segregation Analysis

The variants were analyzed according to their segregation pattern in each family (Appendix A) and were accepted if all patients within the family (with GTS or TD) had at least one alternative allele while none of the non-carriers had it. Only variants that were absent (0%), detected as rare (<1%), or uncommon (<5%) in the control group were further analyzed.

### 4.5. Enrichment Analysis

A functional enrichment analysis was performed to investigate the biological relevance of the genes selected in the above analyses. Lists of genes with detected structural variants co-segregating with the disease were an input to a Metascape software version 3.5.20240101 [75]. Log_10_(q) score <−2 was statistically significant.

## 5. Conclusions

We have identified structural variants and molecular processes related to the etiology of Gilles de la Tourette syndrome in a group of Polish familial cases. The results appear to support conclusions from our previous single-nucleotide variant study associating genes encoding elements of cochlear development and sensory perception of sound with the clinical phenotype of GTS. Also, enrichment analysis provides additional support for the role of synaptic membrane organization, particularly synaptic vesicle endocytosis, and regulation of neuronal migration.

## Figures and Tables

**Table 1 ijms-25-05758-t001:** Rare exonic structural variants co-segregating with the disease in at least one family.

Position (GRCh38)	Type of Variant	Gene	Location	MAF % Controls	MAF % GnomAd	Family, P(obs) ^1^
chr15:44715700-44717603	deletion	*B2M*	partially exon 2, intron 2, exon 3, partially intron 3	0	0	Y, 0.0012
chr19:53410124-53411474	deletion	*ZNF765*	non-coding exon 4, 3′UTR	0	0	I, 0.0024
chr1:215623423-215623500	deletion	*USH2A*	non-coding exon 72, 3′UTR	0.98 (1/102)	0	T, 0.0049
chr18:13647104-13647595	deletion	*LDLRAD4*	within exon 7	0.98 (1/102)	0	J, 0.0097

^1^ P(obs)—probability of cosegregation of an exonic variant in a given family. P(obs) = P(seg_fam) × P(exvar) P(exvar) = 0.0388 as 3.88% of detected variants were exonic.

**Table 2 ijms-25-05758-t002:** Uncommon structural variants co-segregating with the disease in at least two families.

Position (GRCh38)	Type of Variant	Gene	Location	MAF % Controls	MAF % GnomAd	Family, P(obs) ^1^
chr13:113565350-113565735	deletion		intergenic	0	0	A, G, 0.0245
chr15:43793040-43793706	deletion	*SERF2*	intron 2	0	0	G, Y 0.0123
chr2:13096390	55 bp insertion	*ARHGEF4*	intron 5	0	0	G, I, 0.0245
chr2:239642701-239643095	duplication		intergenic	0.98 (1/102)	0	G, I, 0.0208
chr9:86053691	169 bp insertion	*GOLM1*	intron 3	0.98 (1/102)	0	A, G, X 0.0026
chr12:120352608-120352795	deletion	*MSI1*	intron 10	1.96 (2/102)	0	J, R 0.0208
chr14:23717651-23718119	deletion	*ENSG* *00000258464*	intron 3	1.96 (2/102)	0	I, R 0.0104
chr7:1295300-1295559	deletion		intergenic	1.96 (2/102)	0	G, H 0.0833
chr11:69846056-69846309	deletion		intergenic	2.94 (3/102)	0	B, T 0.0078
chr11:76170834	60 bp insertion		intergenic	2.94 (3/102)	0	A, D, I 0.00004
chr16:88401466-88401928	duplication	*ZNF469* ^2^	intron 1	2.94 (3/102)	0	G, I 0.0625
chr1: 231733094	51 bp insertion	*DISC1*	intron 3	3.92 (4/102)	0	D, G, R 0.0005
chr15:55363346-55364831	deletion	*CCPG1*	intron 7	4.90 (5/102)	0	R, X 0.1041
chr18:28521456-28521598	deletion		intergenic	4.90 (5/102)	0	E, G 0.1041
chr4:111086681-111087298-	inversion	*ENSG* *00000288692*	intron	4.90 (5/102)	0	I, X 0.052
chr6:29931995-29933749	deletion		intergenic	4.90 (5/102)	0	G, X 0.2083

^1^ P(obs)—probability of cosegregation of a particular variant in a given families, Bonferroni corrected to possible combinations of families: P(obs) = P(seg_fam1) × P(anyvar) × P(seg_fam2) × P(var) × 136. P(obs) = P(seg_fam1) × P(anyvar) × P(seg_fam2) × P(var) × P(seg_fam3) × P(var) × 680, P(seg) = probability of cosegregation with a disease in particular family, P(anyvar) = 1 − probability of a variant in family 1 with MAF < 0.05. P(var) = MAF − probability of observing the particular variant in another family. ^2^ Another variant in ZNF469, a 71-bp insertion, was cosegregating with a disease in family J.

**Table 3 ijms-25-05758-t003:** Top significantly enriched categories with GTS-associated structural variants.

Term ID	Description	LogP	Genes in Which Variants Were Found:
GO:1900242	**regulation of synaptic vesicle endocytosis**	−5.691	*AGRN*, *ARHGEF4*, *PRKN*, *ROCK1*, *SNX9*, *PTK2*, *RNF139*, *TJP1*, *B2M*, *S100A8*, *CD74*, *ZFYVE28*, *SERF2*
R-HSA-111465	Apoptotic cleavage of cellular proteins	−4.759	*ROCK1*, *PTK2*, *TJP1*, *ITGA8*, *COL4A1*, *TRAK1*, *NRXN3*, *ARHGEF4*, *SNX9*, *PRKN*, *B2M*, *S100A8*, *AGRN*
GO:0031252	cell leading edge	−4.613	*PSD3*, *ARHGEF4*, *DOCK8*, *ITGA8*, *SNX9*, *ROCK1*
GO:0031648	protein destabilization	−4.424	*RNF139*, *PRKN*, *PTK2*, *SNX9*, *S100A8*, *ROCK1*, *SERF2*
R-HSA-375165	**NCAM signaling for neurite outgrowth**	−4.096	*ARHGEF4*, *COL4A1*, *AGRN*, *ITGA8*, *PTK2*, *ROCK1*, *PRKN*
GO:0007611	**learning or memory**	−3.316	*NRXN3*, *ITGA8*, *B2M*, *PRKN*, *TJP1*, *PTK2*, *SNX9*, *ROCK1*, *S100A8*
GO:0045785	positive regulation of cell adhesion	−3.315	*CD74*, *B2M*, *DOCK8*, *PTK2*, *TJP1*, *COPB1*, *SNX9*, *SORCS2*, *PRMT8*, *DNAJC24*, *ROCK1*
GO:0030135	coated vesicle	−3.113	*CD74*, *B2M*, *COPB1*, *SNX9*, *SORCS2*
GO:0043197	**dendritic spine**	−2.825	*SORCS2*, *ITGA8*, *PTK2*, *TRAK1*, *PRKN*
GO:0019904	protein domain specific binding	−2.797	*ARHGEF4*, *PRKN*, *PTK2*, *KHDRBS2*, *TRAK1*
GO:0043484	regulation of RNA splicing	−2.734	*SUPT3H*, *SRRM4*, *KHDRBS2*
GO:2001242	regulation of intrinsic apoptotic signaling pathway	−2.694	*PRKN*, *S100A8*, *CD74*, *AGRN*
R-HSA-9679506	SARS-CoV Infections	−2.672	*B2M*, *TJP1*, *CD74*, *AGRN*, *ROCK1*, *PRMT8*, *DNAJC24*, *PTK2*
GO:0043549	regulation of kinase activity	−2.272	*PTK2*, *SNX9*, *CD74*, *PRKN*, *AGRN*, *TRAK1*

Terms in bold font indicate processes that are neural-specific.

**Table 4 ijms-25-05758-t004:** Top significantly enriched categories with GTS-associated structural variants, based on variants that were found in more than one family, and the results of the overlap with a single-nucleotide variant analysis.

Term ID	Description	LogP	Genes in Which Variants Were Found:
GO:0031256	leading edge membrane	−4.057	*ARHGEF4*, *PSD3*, *USH2A*
GO:0051098	regulation of binding	−3.645	*B2M*, *CCPG1*, *DISC1*
GO:0098793	**presynapse**	−2.611	*DISC1*, *EYS*, *NRXN3*, *USH2A*
GO:0045596	negative regulation of cell differentiation	−2.285	*B2M*, *LDLRAD4*, *USH2A*

Terms in bold font indicate processes that are neural-specific.

**Table 5 ijms-25-05758-t005:** Overview of the study group.

Family Code	Genomes (*n* = 124) [Males/Females]	GTS (*n* = 40)	Non-GTS TD (*n* = 40)	Cosegregation Probability	Healthy (*n* = 44)
A	7 [4/3]	3	1	0.0625	3
B	8 [5/3]	3	2	0.0156	3
C	9 [4/5]	2	4	0.0078	3
D	7 [5/2]	2	2	0.0156	3
E	7 [4/3]	2	2	0.0625	3
F	11 [7/4]	4	4	0.0078	3
G	4 [2/2]	1	2	0.250	1
H	6 [4/2]	2	3	0.125 or 0.125 *	1
I	6 [4/2]	2	2	0.0625	2
J	5 [2/3]	1	2	0.250	2
R	6 [4/2]	1	3	0.125	2
S	6 [2/4]	4	0	0.125	2
T	9 [4/5]	2	3	0.125 or 0.250 *	4
U	6 [5/1]	2	2	0.0625	2
W	14 [7/7]	4	4	0.000976	6
X	5 [2/3]	2	1	0.125	2
Y	8 [4/4]	3	3	0.03125	2

* In families H and T, two segregation patterns were analyzed (Appendix A).

## Data Availability

The data presented in this study are available on request from the corresponding author.

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
