# Peer review of "Structural Variants and Implicated Processes Associated with Familial Tourette Syndrome"

_ijms, 2024, doi:10.3390/ijms25115758_

Round 1

Reviewer 1 Report

Comments and Suggestions for Authors

This is a very good manuscript that undertakes WGS analysis to identify disease-causing genomic variants, focusing on SVs.

As most of the SVa were intronic and the authors struggled to explain the intronic effects, I suggest checking these two articles (PMID: 37511314, 35289213). They explain the pathogenic mechanisms for intronic variants and pseudoexonisation.

Other than that, it is a great paper, and I like that the authors used WGS to do some meaningful genomics. This is an appropriate approach. Kudos to the authors for not using WES that would have been quite inappropriate.

Author Response

Although some copy number variants were already considered as Tourette syndrome-causing, however this is the first, to our knowledge, approach to identify structural variants from whole-genome sequencing data in GTS. While the significance of the detected variants co-segregating with the disease is not clear, we have discussed the possible influence of those variants that are either rare and exonic or co-segregate with the disease in at least two families. In addition, our enrichment analysis has revealed the processes that are most likely to be affected by the detected variants, and we discuss how they might be relevant to the disease.

Obviously, we have compared the obtained results with our previous study on the same group, since structural and single-nucleotide variants do occur in the same group of families. We have presented such a comparison in subsection 2.2, which is further discussed in subsection 3.2.

The description of the possible mechanism of how certain types of variants can affect the gene expression or gene structure was not the scope of our manuscript, therefore we have added only a very brief explanation with appropriate references at the beginning of the discussion. 

Reviewer 2 Report

Comments and Suggestions for Authors

This study analyzes the whole-genome sequencing data previously published by the authors, with a specific focus on identifying structural variants potentially linked to familial GTS. They have identified several gene candidates, along with their known functions, which are potentially associated with familial GTS. This study presents intriguing findings that could offer valuable molecular insights for future research in this area. However, a minor revision of the study is required before it can be further considered for publication in IJMS.

To enhance clarity for readers not well-versed in the field, the authors should provide clarification on how they define uncommon variants, rare variants, and private variants. 

Given that whole-genome sequencing often yields numerous false positive variants, it is crucial for the authors to confirm the existence of these four rare exonic structural variants through sequencing. This verification is essential for researchers interested in potentially exploring the function of these variants in future studies.

Author Response

Although the definition of uncommon, rare, and ultra-rare/private variants is well known to geneticists and the frequency of such variants was mentioned in the manuscript (see subsection 2.1. and 4.4.), we have clarified the meaning of the private variants in subsection 2.1.
Since we performed a high throughput analysis of the structural variants and we did not select any of them as being solely disease-causing, we have omitted this step. The results of the enrichment analysis were based on 211 structural variants co-segregating with the disease. In addition, we have listed 20 of these variants as more interesting (16 co-segregating with the disease in two or three families, and 4 exonic). This is still a large number of variants to be confirmed by Sanger sequencing or PCR followed by agarose gel electrophoresis. However we are confident that the variants detected are true positives. The detection method used is based on a consensus of a few identification algorithms (see pmid: 36315072) and is much more specific than sensitive. It is much more likely to produce false negatives than false positives. Anyway, we have quickly confirmed all variants from table 1. and 2. By examination of aligned sequences (bam files) in the Integrative Genomics Viewer software, which confirmed all 20 variants. Information abut that was added in subsection 4.3. of the manuscript.